# Heterogeneous Evolution of Sex Chromosomes in the Torrent Frog Genus *Amolops*

**DOI:** 10.3390/ijms231911146

**Published:** 2022-09-22

**Authors:** Jun Ping, Yun Xia, Jianghong Ran, Xiaomao Zeng

**Affiliations:** 1Chengdu Institute of Biology, Chinese Academy of Sciences, Chengdu 610041, China; 2Key Laboratory of Bio-Resources and Eco-Environment of Ministry of Education, College of Life Sciences, Sichuan University, Chengdu 610065, China; 3University of Chinese Academy of Sciences, Beijing 100049, China

**Keywords:** sex-chromosome, homomorphy, turnover, X–Y recombination

## Abstract

In sharp contrast to birds and mammals, in numerous cold-blooded vertebrates, sex chromosomes have been described as homomorphic. This sex chromosome homomorphy has been suggested to result from the high turnovers often observed across deeply diverged clades. However, little is known about the tempo and mode of sex chromosome evolution among the most closely related species. Here, we examined the evolution of sex chromosome among nine species of the torrent frog genus *Amolops*. We analyzed male and female GBS and RAD-seq from 182 individuals and performed PCR verification for 176 individuals. We identified signatures of sex chromosomes involving two pairs of chromosomes. We found that sex-chromosome homomorphy results from both turnover and X–Y recombination in the *Amolops* species, which simultaneously exhibits heterogeneous evolution on homologous and non-homologous sex chromosomes. A low turnover rate of non-homologous sex chromosomes exists in these torrent frogs. The ongoing X–Y recombination in homologous sex chromosomes will act as an indispensable force in preventing sex chromosomes from differentiating.

## 1. Introduction

The classic model of sex chromosome evolution suggests that a sex-beneficial mutation occurring close to a newly evolved sex-determining locus is expected to spread. Recombination suppression then expands from a sex-determining locus to larger portions of the sex chromosome [1]. The expansion of the non-recombining region can proceed in a stepwise manner, possibly via chromosomal rearrangements such as inversions [2,3]. The loss of recombination leads to the accumulation of mutations, repetitive elements, and gene loss in sex-limited chromosomes due to the Hill–Robertson effects [4]. Over time, initially identical sex chromosomes are expected to diverge from each other in gene content and nucleotide sequence, and this is thought to have given rise to highly heteromorphic sex chromosomes. This process has been widely approved in older systems, including mammals, birds, and flies [5,6].

However, an increasing number of homomorphic sex chromosomes has been discovered in numerous newly evolved systems, such as in many fishes, amphibians, and non-avian reptiles [7,8,9,10]. In these younger systems, X and Y chromosomes do not show extensive divergence but remain homomorphic [9]. Undoubtedly, recombination suppression not only causes a divergence between the sex chromosomes but also promotes degeneration of the sex-limited chromosome. If repressing recombination continues, extensive differentiation between X and Y chromosomes would be expected. Why does divergence not extend to the sex chromosomes in these younger groups? How is sex chromosome homomorphy maintained here?

One main hypothesis for the homomorphy of sex chromosomes is the view of “high turnover.” This suggests that the ancestral sex chromosomes reverted to autosomes and were replaced by a new set of sex-determining chromosomes. Newly evolved sex-determining genes can convert a pair of autosomal chromosomes into homomorphic sex chromosomes [11]. This explanation predicts that diverged species with homomorphic sex chromosomes have non-homologous genomic locations of their triggers for sex determination. The “high turnover” hypothesis further suggests that the sex-determining locus is regularly replaced by new ones; thus, the non-recombining segments that later expand around the new sex-determining locus do not have enough time to degenerate. Recent turnover events have been observed in several fish [12,13], amphibian [14,15,16,17], and reptile lineages [18]. However, it is not clear whether such events are often sufficient to interpret the overwhelming prevalence of sex chromosome homomorphy. Indeed, most of these animal lineages seemingly show constant heterogametic systems in closely related species, despite shifts (e.g., XY to ZW) that occur occasionally. Moreover, the degree of sequence divergence between gametologs does not always correlate with age [9,19].

Another possible explanation is that homomorphy of sex chromosomes depends on the rate of recombination between them. Homomorphy could be maintained if the recombination frequency between the gametologs was sufficient to prevent divergence between them but was not sufficient to disturb the sex-determining locus [20]. Under these conditions, the genomic location of the trigger for sex determination remains invariant among divergent species. It is supposed that a high rate of recombination might still be in non-sex-linked regions of the genome when the recombination rate is high in the homogametic sex. In this situation, a lower rate of recombination in the heterogametic sex is anticipated by the “X–Y recombination” hypothesis. In many younger lineages, recombination suppression between gametologs appears heterogeneously across the length of the sex chromosomes, suggesting that recombination repression is a more gradual process than those involving series inversions [21,22,23]. If recombination suppression expands gradually, occasional and ongoing recombination between the X and Y chromosomes might persist and stop the degeneration of sex chromosomes [15,19,24,25].

Higher turnover rates may be grossly anticipated when turnover events are frequently reported. However, notably, the values are much more documented for deeply diverged clades than across the most closely related species. Indeed, an in-depth analysis of sex chromosome evolution as an example for very closely related species is limited and currently lacking.

Here, we examined the patterns of sex chromosome evolution in an example of the most closely related species among nine species of the Asian ranid frog genus *Amolops*, a group of torrent frogs. These *Amolops* species were phylogenetically grouped into eastern and western clades, and of the nine species, only one species, *A. wuyiensis*, belonged to the eastern clade, while the remaining eight species belonged to the western clade [26]. The early cytogenetic studies show that all investigated *Amolops* species are homomorphic sex chromosome, except one species (*A. mantzorum*) that has morphologically distinct sex chromosomes [27,28,29,30]. We are interested in, across closely related species of a lineage in a relatively short period of time, whether the turnover of the trigger for sex determination is at a higher rate to prevent sex chromosome degeneration or whether the trigger will remain invariant in the genomic location with the ongoing recombination between sex chromosomes hindering differentiation. In this study, we used the genotyping-by-sequencing (GBS) approach to search for sex determination systems and the identity of sex chromosomes in nine species. We reconstructed a multispecies gene tree to further identify signatures of recurrent X–Y recombination [31,32,33] to estimate the forces that prevent sex chromosomes from degeneration over time. Our results show that the sex-chromosome homomorphy results from both turnover and X–Y recombination in the torrent frog genus *Amolops*, simultaneously representing heterogeneous evolution on homologous and non-homologous sex chromosomes. In contrast with deeply diverged clades, the *Amolops* species appears to have a lower turnover rate of non-homologous sex chromosomes. Moreover, across these most closely related species of *Amolops*, the ongoing X–Y recombination in homologous sex chromosomes would act as an indispensable force to prevent the sex chromosomes from extensive divergence.

## 2. Results

### 2.1. Simplified Genomic Data Analyses

GBS libraries were created for all nine species, in which the data of *A. mantzorum* were combined with those published previously by Luo et al. (2020) for analysis [34]. A total of 354.28 Gb (Gbase) GBS raw data were obtained for the nine species datasets, ranging from 21.734 to 61.316 Gb for each species. With regard to the RAD-seq library constructed for *A. jinjiangensis* and *Amolops* sp., a total of 168.567 Gb of raw data were obtained, ranging from 73.922 to 94.645 Gb for each species. For more information on the simplified genome data for each species, see Appendix A. Clean data after demultiplexing and filtering in the *process_radtags* module were used for subsequent assembly analysis (Appendix A).

To obtain more loci to avoid merging real sex-linked loci, two standards for assembly analyses were employed for each GBS dataset. One uses optimal parameters that are obtained by monitoring the samples prior to analysis (Appendix A), and the other uses the lower ones (−M = 2, −n = 1, −m = 2). For the RAD-seq dataset, only the lower parameters (−M = 2, −n = 1, and −m = 2) were used. With the optimal parameters, nine catalog files containing 23,448,367 loci were generated for the GBS datasets of the nine species, ranging from 1,581,852 to 3,944,880, which contained 265,349 to 1,162,812 polymorphic loci. Under the parameters (−M = 2, −n = 1, and −m = 2), the nine catalog files contained 46,893,516 loci, and ranged from 3,206,844 to 7,309,508, which contained 437,564 to 1,699,515 polymorphic loci. The catalog file for RAD-seq of *A. jinjiangensis* contained 14,187,882 loci (795,492 loci were polymorphic), and that of *A.* sp. contained 16,365,648 loci (2,183,830 loci were polymorphic). When the nine-species unified dataset was analyzed, catalog files contained 33,949,023 loci, which contained 11,797,189 polymorphic loci.

### 2.2. Identifying Sex-Linked Markers and Sex-Determination Systems

We used three complementary approaches to identify sex-linked markers, respectively, based on (i) sex differences in allele frequencies, (ii) sex differences in heterozygosity, and (iii) sex-limited occurrence. Our screens for putative sex-linked markers using the three strategies yielded signatures that matched our expectations for either an XY or a ZW system in all datasets (Appendix A), indicating the prevalence of false positives. Therefore, opposite sex verification was used as an in silico approach to test whether the observed numbers of putative XY or ZW markers differed from random expectations. Then, for all nine species, the markers expected by the ZW system were excluded, and we found that all species investigated here were from XY sex determination systems. 

We subsequently combined all sex-linked markers for each individual from the two GBS datasets analyzed with the different standard assemblies and RAD-seq dataset by removing duplicate markers. The final number of sex-linked markers identified in each dataset varied widely, ranging from 35 to 343 (Appendix A), and only these markers were used in the analysis. 

As with the SNP-based analysis of the nine-species unified dataset, we found no male-specific GBS tag shared across all nine species, except one between an alternate species pair, that is, *A. xinduqiao* and *A.* sp. (Appendix A).

Of these markers, several were randomly selected for PCR verification in all species except *A. mantzorum*, which has been previously validated [34]. All primers used are listed in Appendix A. After amplification of samples of both males and females from each species, the PCR products were sequenced at the Sangong Sequencing Center (Shanghai, China). All male individuals were heterozygous, while all females were homozygous for these polymorphic loci (Appendix A). Thus, the XY sex determination system was further confirmed by PCR amplification in all nine *Amolops* species (Appendix A). 

### 2.3. Identifying the Sex Chromosome

Using sex-linked markers, the genomic positions of the sex chromosomes could be determined. To determine whether sex chromosome conversion occurred among the *Amolops* species based on the position of the sex-linked locus on the chromosomal genome, the reference genome of closely related species that was used for whole-genome sequencing was selected as a third-party genome. Sex-linked markers were located to determine whether the turnover of sex chromosome pairs occurred and to determine the type of turnover [35,36].

At present, neither the whole-genome sequence nor a sketch of the whole genome has been obtained for *Amolops* species, and thus, genomic data of these species should be compared with the genome data of other closely related species. Sex chromosomes were identified by mapping the confirmed sex-linked markers onto the reference genome by direct and indirect methods. Firstly, we tried to blast the confirmed sex-linked markers directly to the reference genome of closely related species. Using the direct mapping method, the results showed that the sex-linked markers of the nine *Amolops* species were located and dispersed in the reference genomes of *Pyxicephalus adspersus*, *Rana temporaria*, and *Rhacophorus Kio*. The sex chromosomes could not be determined. More than 91% of the sex-linked markers failed to be successfully aligned to the *P. adspersus* genome, more than 41% of the sex-linked markers failed to be successfully aligned to the *R. temporaria* genome, and more than 71% of the sex-linked markers failed to be successfully aligned to the *Rh. kio* genome (Appendix A).

Secondly, we devised a two-step strategy by first mapping our confirmed sex-linked markers to a draft *Lithobates catesbeianus* genome, then extracted 2 kb of the *L**. catesbeianus* scaffold each side of the hit, and blasted these 4 kb sequences to the *P**. adspersus* genome [35,37,38]. By this indirect mapping, we compared the sex-linked markers of all nine *Amolops* species to the *P. adspersus* genome. The results showed that the sex-linked markers of nine *Amolops* species were scattered in the genome of *P. adspersus* and were not significantly concentrated on one chromosome; furthermore, more than 68% of the sex markers failed to be successfully aligned (Appendix A).

Therefore, none of the alignments showed significant concentrations on one chromosome when the three species *P. adspersus* from Pyxicephalidae, *Rh. kio* from Rhacophoridae, and *R. temporaria**,* even from the same family of Ranidae, were used as the third-party reference genome. The reasons for the unsuccessful alignments of sex-linked markers were the high sequence divergence between the *Amolops* species and the third reference frame. This high sequence divergence could be because these sex-linked markers are non-coding genes with high differentiation or no homologous sequence in reference genome. Such differentiation could be evolved in relatively recent evolutionary time (that is, *R. temporaria*, approx. 40 M years; *Rh. Kio*, approx. 70 M years; *P. adspersus*, approx. 80 M years) [39].

Finally, the sex-linked markers of eight of the nine *Amolops* species were successfully aligned to the genome of the sex chromosome (No. 5) of *A. mantzorum*, which was previously shown to be homologous among the *A. mantzorum* species group [40,41]. The results showed that, except for *A. wuyiensis*, of which only 13.8% of the sex-linked markers were successfully aligned, the majority of the sex-linked markers, ranging from 41.5% to 71.4%, were successfully mapped (Appendix A). Accordingly, the sex-linked chromosomes of eight of the nine species (exception: *A. wuyiensis*), could be identified as homologous to chromosome 5 in *A. mantzorum*. 

### 2.4. Identifying Potential Sex-Determining Genes

Of the nine species, five species had several sex-linked markers that were separately mapped on only two candidate genes for sex determination in frogs (Figure 1; Appendix A). Among them, nine sex-linked markers of *A. mantzorum* were located in the *CYP19A1* gene of *R. rugosa*, and two sex-linked loci were consistent with those identified by Luo et al. (2020) [34]. *A. xinduqiao* had four sex-linked loci, whereas *A. jinjiangensis* and *A. tuberodepressus* had one sex-linked marker that was mapped to the *R. rugosa CYP19A1* gene. In addition, *A. wuyiensis* has a sex-linked marker located in the *R. rugosa AR* gene. Appendix A shows the matching nucleotide alignment of the *BLASTn* hit results of sex-linked loci of various species sharing more than 77% identity. The GenBank accession numbers of the hits are shown in parentheses. “Start” and “End” indicate the nucleotide positions of the *CYP19A1* and *AR* genes. Notably, in *R. rugosa*, it was previously shown that *AR* is located on chromosome 7 (i.e., sex chromosome) [42], whereas *Cyp19A1* is located on chromosome 3 [43,44]. This indicates that these two genes are located on different chromosomes rather than on identical ones in ranid frog *R. rugosa*.

### 2.5. Patterns of Sex-Chromosome Turnovers

We directly placed the sex chromosome identities of each species in the phylogenetic tree, as shown in Figure 1. The sex chromosome is mapped with the sex-linked markers in each of the nine *Amolops* species, and there is no shared XY site (i.e., sex-linked locus) except one between alternate species pairs of *A. xinduqiao* and *A.* sp. (Appendix A). The results showed that two pairs of sex chromosomes existed, each carrying a different sex-determining candidate gene (*CYP19A1* or *AR*). This allowed us to infer that one sex chromosome turnover event occurred between the western and eastern clades in the tree within 31 million years [26].

### 2.6. Gene Trees of X–Y Recombination

The approach for identifying signatures of recurrent X–Y recombination in *Amolops* species uses multispecies gene trees. Following the rationale of Stöck et al. (2011) [19], the X and Y alleles are expected to cluster by gametologs in the absence of sex chromosome recombination but by species otherwise. Two species were selected for the simulation and deduction of the multispecies gene trees (Figure 2). If the sex-linked region on the Y chromosome originated from a common ancestor of the two species, that is, sex chromosome differentiation occurred before species divergence, and the recombination inhibition between X and Y was not disturbed for a long time, then this region of the Y chromosome between species would be more closely related to the X chromosome of their own species (Figure 2a). Conversely, if the sex-linked regions in species one remained and the regions in species two were no longer sex-linked owing to recombination, then the sex-linked regions of X and Y within each species would be more closely related than their homologous sequences in other species (Figure 2b).

Multispecies gene trees were constructed using sex-linked loci of a certain species and cross-species homologous sequences of other species, where one locus or five loci for a species were selected for tree construction. 

With the SNP-based analysis of the nine-species unified dataset, one shared sex-linked locus was found only between a species pair of *A. xinduqiao* and *A.* sp. All male genotypes for this sex-linked locus were T/A heterozygous, whereas all female genotypes were T/T homozygous, which confirmed that the marker was gender-diagnostic in both species (Appendix A). Using this site to construct a multispecies gene tree, both maximum likelihood (ML) and neighbor-joining (NJ) methods generate the same topological tree structure (Figure 3), which is consistent with the prediction in Figure 2a, the X and Y alleles were clustered by gametologs due to the absence of sex chromosome recombination.

As shown in Figure 4, the results were in line with the predictions shown in Figure 2b, which depended on species rather than on gametologs. All the X-specific sequences of *A. chunganensis* were clustered together and apart from the Y-specific sequences of this species when using the five sex-linked loci identified from *A. chunganensis*. In contrast, the X- and Y-specific sequences cannot be identified by using the cross-species homologous sequences of the five sex-linked loci from *A. chunganensis*. In this case, the cross-species homologous sequences of the remaining species clustered according to species rather than gametologs. This indicated that the homologous sequences, which homology to the sex-linked loci identified from *A. chunganensis*, in the remaining seven species no longer sex-linked form. This result was consistent with that of the single-locus tree (Appendix A).

For the remaining seven species with identical sex chromosomes, the gene trees with both one locus and five loci showed topological structures similar to those of *A. chunganensis* (Appendix A). All sex-linked markers and their cross-species homologous sequences used to construct the maximum likelihood tree presented in Appendix A.

## 3. Discussion

### 3.1. Turnover between Non-Homologous Sex Chromosomes

All nine species showed male heterogamety (XY), and there was no shift in the patterns of heterogamety found. Furthermore, all species shared the same sex-linked chromosome pair except one, *A. wuyiensis*, which was displayed on the other. As the sex chromosomes of western and eastern clades harbored different candidate genes (*CYP19A1* or *AR*, respectively) for sex determination, the sex determination pathway probably changed between the two clades (Figure 1). Thus, only one turnover event was detected between the two pairs of non-homologous sex chromosomes among the nine species. Phylogenetically, this turnover event occurred between the eastern and western clades of the genus *Amolops*, which diverged approximately 31 M years ago [26]. This rate of turnover is relatively low across the most closed species in the genus *Amolops*.

Across closed species of a lineage restricted in a geographical region, a lower rate of turnover between non-homologous sex chromosomes was recorded in most cases of frogs that have been investigated. In the African clawed frog clade of the subgenus *Xenopus*, the turnover event was observed only once among 24 species within approximately 50 M years, in which the sex-determining gene *DM-W* (*DM* domain-containing W-link) was detected as the most recent common ancestor of all species [46]. In addition, one turnover (ZW to XY transition) occurred in the European tree frog clade, which included four species of the *H. arborea* group as well as the Tyrrhenian *H. sarda* and the Middle Eastern *H. savignyi* within approximately 11 M years [15,17]. Jeffries et al. (2018) [35] pointed out the fast rate of turnovers in some ranid frogs, in which these frogs diverged deeply into five clades. In fact, a lower rate of turnover was still observed within each of the five clades, except for the American *Rana* clade and the species *Glandirana rugosa*. Only one turnover event was found in the genus *Pelophylax* within approximately 40 M years, whereas it occurred twice in the European *Rana* species and Asian *Rana* species clade within approximately 20 M years, respectively. Sex chromosomes are conserved in the cichlid fish genus *Tropheus*, although a higher rate of turnover is found in other cichlid tribes [47]. The sex chromosomes might be shared in the endemic clad of East Asia Cyprinidae fish and maintained in all diploid fish in the group for approximately 9 M years [48]. 

Practically, despite very few examples, the rapid turnover rates of sex chromosomes have been measured on a large evolutionary scale of taxa that comprise deeply diverged clades, such as those described in true frogs [35] and cichlid fish [47]. Accordingly, when estimated on a small scale across most closed species in a single clade, the rate of turnover would be lower or sex chromosomes would be conserved with no turnovers occurring.

### 3.2. X–Y Recombination Evolved in Homologous Sex Chromosomes

Chromosome 5 was found to be the sex chromosome in all eight species as an XY system and was thus chosen for sex determination only within the western clade of *Amolops* species detected in the present study (Figure 1) [26]. This genomic region may result from the presence of important genes in the sex determination cascade. In the present study, a strong candidate might be *CYP19A1*, a gene involved in several sex-linkage markers in at least four out of the eight species (Figure 1; Appendix A). *CYP19A1* is highly conserved in vertebrates as the sexual differentiation gene and plays a crucial role in gonadal differentiation and development [49,50,51]. Dimorphic expression of *CYP19A1* has been observed in several anuran species during ovarian differentiation, such as in *R. rugosa* and *X. laevis*, indicating that this gene is closely implicated in ovarian differentiation in frogs [52,53,54]. Moreover, there are no other genes associated with sex-linkage SNP markers, except for *CYP19A1*, which was thus assumed to be shared as a candidate for sex determination in western clade species. In addition, the majority of sex-linked markers was successfully mapped to the western clade species (Appendix A). Sex chromosomes are conserved in the western clade of closely related torrent frogs, where the trigger gene for sex determination is assumed to be conserved and remains invariant in at least four out of eight species within the western clade of *Amolops* species. 

In gene genealogies, the “X–Y recombination” hypothesis predicts the identification of an ancestral sex-determining region (SDR) across diverse species [55]. This further suggests that the phylogenies for the sex-linked genes on the ancestral SDR should be clustered by gametologs (Figure 2a). In this study, only one XY site was detected and shared between two species, *A. xinduqiao* and *A.* sp. (Appendix A). At this locus, the sequences were clustered by gametologs rather than by species, suggesting that recombination suppression was finalized before the split of these two species (Figure 3). Despite this locus, no male-specific tags were shared across all eight species in this study (Appendix A). This ancestral SDR may be very small across diverged species. In hylid frog *H. arborea*, it is limited to a 0.5 Mbp interval harboring only three genes, i.e., *DMRT1*, *DMRT3*, and *DMRT2* [55]. Thus, it is possible that we underestimated the number of loci predicted for an ancestral SDR by assigning sex-specific markers via the GBS approach. Accordingly, a genome survey with GBS markers failed to detect small trans-species SDR across four European hylid species [55]. Using similar GBS approaches, the sex-linked markers in this study might have also failed to identify the small ancestral SDR among the eight torrent frogs.

Despite the genes on the ancestral SDR, the “X–Y recombination” hypothesis further predicts the reverse pattern, where the phylogenies for all sex-linked genes would cluster by species instead of by gametologs (Figure 2b). Outside this limited region, the multispecies gene trees revealed that all sex-linked homologous sequences were always clustered by species across all eight species (Figure 4; Appendix A), indicating that X and Y should have continued to recombine. X–Y recombination has been investigated in bufonid toads [24]. It has also been shown in hylid frogs, not only across closely related species of the *H. arborea* clade [19] but also in the deeply diverged *H. meridionalis* and East Asian *H. suweonensis*, where it even involves Z–W recombination [15]. Similarly, in guppy fishes, X–Y recombination has been documented in *Poecilia reticulata* and *P. wingei*, although it has been difficult to quantify the proportion of recombination nodules that result in strand invasion. The fact that the guppy sex chromosome system is at least 20 M years old suggests that X–Y recombination between sex chromosomes can remain long-term [33]. Across three *Gasterosteus* stickleback fishes, the recombination between X and Y chromosomes has been evidenced by the species gene trees that are constructed with sex-linked homologous sequences, whereas the homologous sequences can even evolve striking differences, indicating that the recombination suppression between gametologs appears heterogeneously across the length of the sex chromosomes [56]. 

Across closely related species, sex chromosomes can be conserved by bearing the same trigger gene for sex determination. Thus, the ongoing X–Y recombination between sex chromosomes acts as an indispensable force to prevent extensive divergence of sex chromosomes.

## 4. Materials and Methods

### 4.1. Sample Collection

In total, we generated GBS and RAD-seq data for nine species of the torrent frog genus *Amolops* and re-examined the previously published species *A. mantzorum* [34]. We obtained male and female samples from each species (mean number of females = 10.1, mean number of males = 10.1) for GBS and RAD-seq, leading to a total of 182 individuals (Appendix A). A total of 176 individuals from nine species was used for PCR verification (Appendix A). 

The phenotypic sex of the adults was determined by observing the secondary sexual characteristics and anatomical structures (testes or ovaries) after euthanasia. Liver and muscle tissues were stored in 95% alcohol at −20 °C. All animal procedures were approved by the Animal Protection and Utilization Committee of the Chengdu Institute of Biology, Chinese Academy of Sciences (license number: CIB-20121220A, CIBDWLL2021001).

### 4.2. Genomic Library Preparation

Genomic DNA was extracted separately from the muscle tissues of nine species of *Amolops* (n = 182, Appendix A) using a DNeasy Blood and Tissue Kit (Qiagen), according to the manufacturer’s instructions for animal tissues. The extracted DNA was quantified, and the quality was checked using a nanophotometer spectrophotometer (Implen, West Lake Village, CA, USA) and a qubit DNA assay kit in a Qubit 2.0 fluorometer (Life Technologies, Carlsbad, CA, USA). Genotyping-by-sequencing (GBS) was used to build a genomic library for all nine species [57]. Briefly, the DNA of each individual was digested with *MseI* (New England Biolabs, Ipswich, MA, USA). Then, a 4–8 bp P1 and P2 adapter (complementary to the cleavage DNA gap) was connected to both ends of each individual’s restriction fragment to identify individuals after sequencing. The tag sequence fragments containing P1 and P2 linkers at both ends of each individual were amplified using PCR, fragments of 350–500 bp were collected using agarose electrophoresis and paired-end 150 bp high-throughput sequencing was performed using the Illumina NovaSeq 6000 sequencing platform. 

Additionally, restriction site-associated DNA sequencing (RAD-seq) libraries were constructed for two species, *A. jinjiangensis* and *Amolops* sp. Genomic DNA from each sample was digested with *EcoRI*, and an adapter (P1) was ligated to the compatible end of the fragment. This adapter contained forward amplification and Illumina sequencing primer sites, as well as a 6-bp-long nucleotide barcode for sample identification. Adapter-ligated fragments were then pooled, randomly sheared, and size-selected. The DNA was then ligated to a second adapter (P2), which was a Y-shaped adapter with a different end. The method for RAD-seq is detailed by Baird et al. (2008) [58]. Fragments of 200–400 bp and 400–600 bp were collected for library construction. The quality of the libraries was analyzed using Qubit 2.0 kit. After diluting the library to 1 ng/µL, Agilent 2100 was used to determine the insert size of the library. When the insert size was appropriate, qPCR was performed to determine the effective concentration of the library (>2 nM). All steps were performed to ensure library quality. The constructed libraries were sequenced using the IlluminaHiSeq2500 platform, generating 125-bp paired reads. The genomic library building process was performed at Beijing Novogene Bioinformatics Technology Co. Ltd., Beijing, China (www.novogene.cn (accessed on 3 August 2020)).

### 4.3. De Novo Assembly and SNP Calling

The *process_radtags* module in Stacks-2.41 was then used to demultiplex and filter raw reads. In particular, low-quality sequences and N-containing sequences were filtered out, and the restriction sites were checked and corrected [59,60,61]. The clean data filtered by the *process_**radtag**s* module were then spliced and assembled. As the research object selected in this study did not have a suitable reference genome, a follow-up analysis was completed without a reference genome. The assembly process used the *denovo_map.pl* pipeline in Stacks-2.41. Finally, a *population* program was called to obtain the results [61]. 

As the optimal parameter values of different datasets depend on the genetic diversity and characteristics of the original sequencing data, it is necessary to optimize the small-sample parameters to obtain more data information before assembling large-sample *denovo_map.pl* components [61]. This involves running a series of de novo analyses with different parameter values and monitoring the number of polymorphic GBS found in 80% or more samples (*population* parameter: −r = 0.8) until an optimal set of parameters (M, n, and m) is found. The specific parameter optimization method was described by Paris et al. (2017) [62]. Finally, the Stacks *denovo_map.pl* pipeline was applied to the full dataset using optimal parameters.

However, it is generally impossible to demonstrate, in absolute terms, that a specific parameter combination is optimal. The goal was to fully understand the dataset so that reasonable choices could be made [61,63]. To obtain as many loci as possible, M = 2, n = 1, and m = 2 were simultaneously specified for *denovo_map.pl* pipeline analysis. M, n, and m were set too low because it was considered more important to obtain as many loci as possible than to eliminate false reads, as these are unlikely to pass the strict downstream filtering standards [63]. 

In addition, to test whether there was a shared sex-linked marker among the species of *Amolops*, data from nine species of *Amolops* were obtained, and unified component analysis of the *denovo_map.pl* pipeline was performed (parameters: −M = 2, −n = 1, and −m = 2). If there were consistent sex-linked sites among the species of *Amolops*, they were merged [64]. To control for the potential noise produced by population structure [35], the nine-species unified dataset analysis was only used to identify whether there were shared sites among species.

### 4.4. Filtering for Sex-Linked Markers

After analysis using the *denovo_map.pl* pipeline, the results were output through the *population* program in Stacks-2.41. The *populations.sumstats.tsv* file was used to identify the putative sex-linked SNP loci. Similarly, the match between each individual and catalog was calculated in the *matches.tsv* file to determine the locus depth of each individual, and the loci that existed in all individuals of one sex and not in the other were screened out, which are called putative sex-limited GBS/RAD-seq loci. All of the above were calculated using a custom R script, and three strategies were used to screen gender-related molecular markers [37,63]. The process can be explained by taking the male heterozygous subsystem (XY) as an example and the female heterozygous subsystem (ZW) as the opposite; these approaches are described briefly below. For simplicity, we only describe the changes in the method. (i) Sex-linked SNPs were screened based on their frequency. (ii) Sex-linked SNPs were screened based on heterozygosity. For methods (i) and (ii), the site could be considered a putative sex-linked SNP marker, as it was homozygous in at least one-third of females and heterozygous in at least one-third of the male samples. (iii) Sex-limited loci are used in the third strategy. Thus, they are considered sex-linked if they are completely absent in homogamous sex (XX) and present in at least half of heterozygous sex.

### 4.5. Validating Sex-Linked Markers

False positives in the screened putative sex-linked markers were removed [18,35,63,65]. The following methods were used to verify that real and effective sex-linked markers were obtained: (i) Verification of opposite sex was used to validate the sex-linked markers, which was confirmed with BLAST and grep searches [63]. Considering the XY sex determination system as an example, the ZW system had the opposite effect. For example, Y-linked SNP markers have been identified. First, the GBS/RAD-seq clean data of all individuals in each species were divided into two datasets according to sex, and the files in the male and female folders were merged into one file. Local databases for females and males were established using BLAST 2.6.0+. The sex-linked SNP markers were aligned to female and male databases using local *BLASTn* (E-value ≤ 1 × 10^−20^). If there was no meaningful alignment of the variable base of Y-linked SNP markers in the female database and the results were consistent in the male database, the sex-linked loci were retained; otherwise, it was a false-positive locus. For sex-limited markers, the GBS/RAD-seq clean data of all individuals in each species were divided into two datasets according to sex. Using the Linux grep command, for example, the Y-limited loci obtained by preliminary screening were compared with the female and male GBS/RAD-seq datasets. If Y-limited loci did not have the same sequence in female GBS/RAD-seq datasets, and at the same time, if the Y-specific site could find the same sequence in the male GBS/RAD-seq dataset, it was an effective Y-specific marker; all other sites were rejected as false-positive sites. All loci that passed this stage were called “confirmed” sex-linked markers. (ii) PCR verification was further performed to validate sex-linked markers. To verify that the gender-linked markers were real and effective sex-linked markers and not just for individuals in GBS/RAD-seq, PCR primers were designed to amplify additional sample DNA. For the specific methods, refer to Luo et al. (2020) [34].

### 4.6. Assigning Sex-Linked Markers onto Sex-Chromosome

Comparative genomic studies have consistently demonstrated extensive collinearity between anuran-born chromosomes, except for occasional differences in chromosome numbers due to fusion or fission [37,55,66,67,68]. Based on the distance between genetic relationships, the sex chromosome (No. 5) genome of *A. mantzorum* [34] and the genomes of *Rana temporaria*, *Pyxicephalus adspersus* [38], and *Rhacophorus kio* [69] were selected as the third-party reference frame. BLAST 2.6.0+ was used to map sex-linked loci onto the reference genome using *BLASTn* to determine the chromosomal location of sex-linked loci in the third-party reference genome (matches were retained only if their E-values were ≤1 × 10^−20^. In the case of multiple matches, at least five orders of magnitude lower than the next-best match). Positioning methods are divided into indirect and direct mapping methods. The third-party reference genome of *P. adspersus* was used as an indirect location [35,37]. The sex-linked markers obtained were aligned with the scaffolds of *Lithobates catesbeianus* using *BLASTn*. The length of 2 kb was intercepted before and after the alignment position using Python script, the extended fragment was aligned to the reference genome of *P. adspersus*, and the relative position of sex-specific markers on the third-party reference genome was determined. Direct mapping method: The sex-linked loci were directly mapped to the reference genome using *BLASTn* to determine the relative positions of sex-linked markers on the third-party reference genome. Direct mapping methods were performed using chromosome 5 of *A. mantzorum* [34], *P. adspersus* [38], *R. temporaria*, and *Rh. kio* [69], as a third-party reference genome.

### 4.7. Predicting the Sex-Linked Markers Involved in Genes for Sex Determination

The molecular mechanisms of sex determination in amphibians are unknown, partly because a pair of sex chromosomes is difficult to find [70]. According to the results of the sex-linked gene and karyotype data, at least eight genes were selected as candidate genes for sex determination in frogs, including *AMH*, *AR*, *CYP17*, *CYP19A1*, *DMRT1*, *FOXL2*, *SOX3*, and *SF1* [10,70]. At present, the female sex-determining gene identified in amphibians is the *DM* domain-containing W-link (*DM-W*), and it was found in *Xenopus laevis* [71], a species with an allotetraploid origin and homomorphic ZZ/ZW sex chromosomes [72]. This gene is a deletion mutant of *DMRT1* that lacks a transactivation domain [73]. To verify whether there was an association between the obtained sex-linked markers and sex determination candidate genes, the sex-linked markers in the NCBI nucleotide database were directly searched using *BLASTn* (https://blast.ncbi.nlm.nih.gov/Blast.cgi (accessed on 8 May 2021)) with the goal of obtaining their potential functional information [34,63,74]. 

### 4.8. Assessing Patterns of Sex-Chromosome Turnover

To assess the pattern of sex chromosome turnover or whether some chromosomes are more likely to be recruited, it was necessary to put all the sex chromosome identities of *Amolops* species into a phylogenetic tree [35]. The phylogeny of the genus *Amolops* has been well resolved [26,45]. Thus, phylogeny was generated using a combination of data from previous phylogenetic studies. All sex chromosome identities of the *Amolops* species investigated here were directly included in the phylogenetic tree. 

### 4.9. Multispecies Gene Trees

Multispecies gene trees were constructed using sex-linked SNP markers of a certain species and cross-species homologous sequences of other species. First, a local library was constructed using the *makeblastdb* command in BLAST 2.6.0+ to construct the *populations.samples.fa* file of each species (the FASTA file of two haplotype sequences of each diploid sample for each locus). Subsequently, the sequence alignment was completed using the *blastn* command, and the homologous sequence of a sex-linked locus for one species was subsequently selected for the other, and only the alignment result of E-value ≤ 1 × 10^−20^ was retained [17,56]. The sequences were aligned using SeaView v5.0.5 (https://doua.prabi.fr/software/seaview (accessed on 6 March 2022)). Finally, the maximum likelihood evolutionary tree was generated using RAxML v.8.2.11 [75] under the GTR + G model and fast bootstrap analysis with more than 1000 guides (−f = a). MEGA v. 7.0 [76] was used to create a neighbor-joining (NJ) tree based on the Kimura-2-parameter model with 10,000 bootstraps. View trees were constructed using FIGTREE v.1.4.4 (http://tree.bio.ed.ac.uk/software/FigTree/ (accessed on 22 March 2022)). 

## 5. Conclusions

In conclusion, we analyzed the patterns of sex chromosome evolution among nine species of the torrent frog genus *Amolops*. The XY sex determination systems were investigated in all nine species by performing simplified genomic sequencing and verifying the findings using PCR. In total, 35 to 343 confirmed sex-linked markers were obtained in each species data set. We found that both turnover and X–Y recombination are responsible for the sex-chromosome homomorphy in *Amolops.* The non-homologous sex chromosomes proved that one turnover event exists between western and eastern clades but without shift of the patterns of heterogamety. Furthermore, we found that across closely related species, sex chromosomes can be conserved by bearing the same trigger gene for sex determination. Our results show that X–Y recombination is ongoing between sex chromosomes and is an indispensable force that prevents extensive divergence of sex chromosomes.

## Figures and Tables

**Figure 1 ijms-23-11146-f001:**
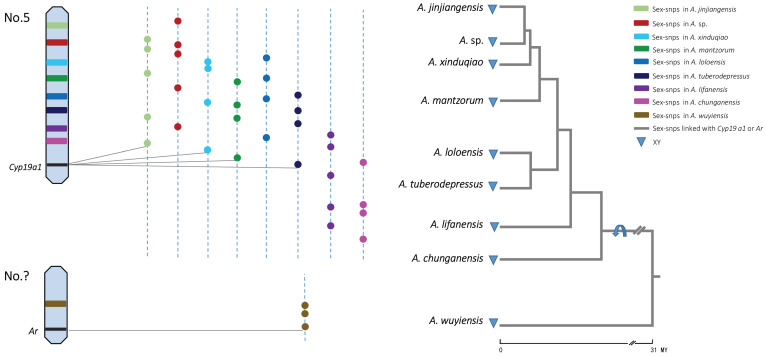
Sex chromosome turnover across nine torrent frog species. Sex-determination system and sex-chromosome identities come from both GBS and RAD-seq (see detail in Appendix A) data. Two chromosome pairs harbor a different candidate gene for sex determination: one is chromosome 5 and the other is unknown. Blue arrow shows the branch on which the inferred turnover occurs based on the direct mapping analyses. The dotted lines represent the sex chromosome for each species, and colored dots show sex-linked SNPs of nine *Amolops* species. Sex-linked SNPs are randomly located and there is no shared XY site except one between alternate species pairs of *A. xinduqiao* and *A.* sp (the site is not shown here). The grey solid lines show some sex-linked SNPs for five species were mapped to *Cyp19a1* and *Ar* genes, respectively. The tree topology was adapted using a combination of data from mitochondrial phylogenies by Lu et al. (2014) [45] and Zeng et al. (2020) [26], and approximate divergence times (million years, My) were generated from molecular dating by Zeng et al. (2020) [26].

**Figure 2 ijms-23-11146-f002:**
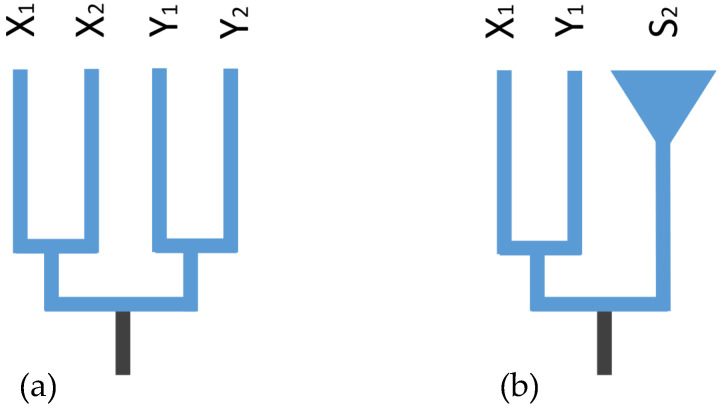
Gene genealogies under different evolutionary scenarios. (**a**) The marker is on the ancestral sex-determining region of sex chromosomes and thus is sex-linked in two species, but due to the absence of X–Y recombination, alleles cluster according to gametologs, and not species. (**b**) The marker is on sex chromosomes but outside the ancestral sex-determining region and thus sex-linked in species one rather than species two, whose genealogies conform to species genealogy when X–Y recombination occurs in the latter.

**Figure 3 ijms-23-11146-f003:**
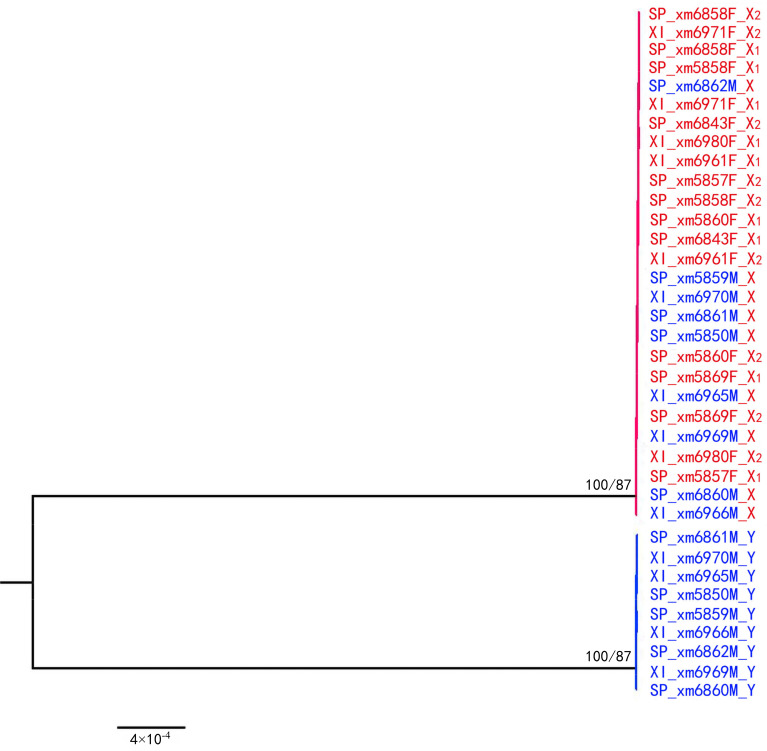
Gene genealogies for the shared XY site. Sequence label feature species (XI for *A. xinduqio*; SP for *A.* sp.), sex (M for male; F for female), ID number, and allele (X, Y). Sequences from males are shown in blue, and those from females in red. The markers are on the ancestral sex-determining region of the sex chromosomes, and thus, are sex-linked in both species. Due to the absence of X–Y recombination, all sequences cluster by gametologs not by species. Bootstrap values are shown for the main branches (ML/NJ), when above 50%.

**Figure 4 ijms-23-11146-f004:**
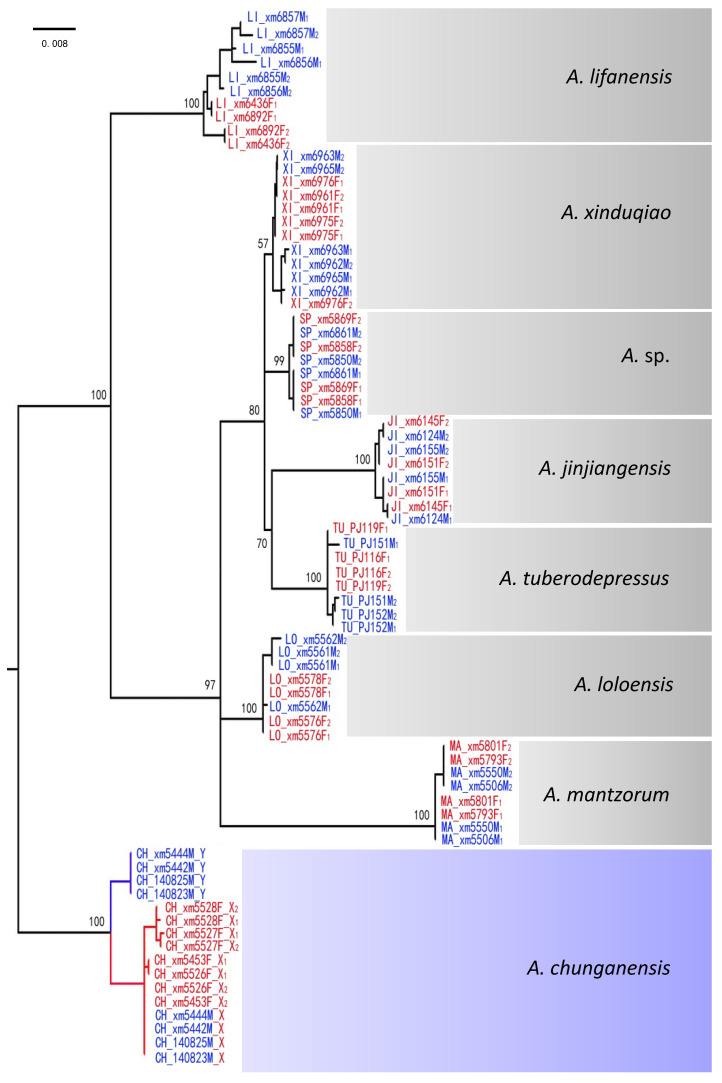
Gene genealogies of sex-linked loci outside ancestral SDR. Sequence labels of feature species (CH for *A. chunganensis*; JI for *A. jinjiangensis*; LI for *A. lifanensis*; LO for *A. loloensis*; MA for *A. mantzorum*; SP for *A.* sp.; TU for *A*. *tuberodepressus*; XI for *A. xinduqio*), sex (M for male; F for female), ID number, and allele (X, Y). Sequences from male individuals are shown in blue, and those from females are shown in red. The markers are on sex chromosomes outside the ancestral sex-determining region (SDR) and thus sex-linked in *A. chunganensis* and not in the remaining seven species. All sequences group together by species genealogy, while X–Y recombination occurs in all the remaining species. Bootstrap values are shown for the main branches, when above 50%.

## Data Availability

All GBS and RAD-seq data presented in this study can be found in the Sequence Read Archive (SRA) database under the accession code PRJNA870957. Stack outputs, R intermediate files, and PCR sequences were archived in Dryad (https://doi.org/10.5061/dryad.b2rbnzsjn (accessed on 8 September 2022)).

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
