# Peer review of "Heterogeneous Evolution of Sex Chromosomes in the Torrent Frog Genus Amolops"

_ijms, 2022, doi:10.3390/ijms231911146_

Round 1

Reviewer 1 Report

For someone not working directly in the field, I found the manuscript from Jun Ping and colleagues interesting but not always easy to read. The method section contains a lot of information that is useful for understanding how the authors did their analysis, but also information that could (should) be used in the results part to help the reader understanding the logic/rational behind each type of analysis. A part of this information should be transferred from the methods to the main text. Also the authors should better highlight the conclusion(s) of their work and what it does bring to the field they are working in.

Specific points:

Introduction

Page2: "might still be enjoyed". Not sure about the use of "enjoyed" here.

Page 2: "the lower turnover rate of non-homologous sex chromosomes exhibits in Amolops species". Something wrong I think with the use of "exhibits" in this sentence. Should be turned around.

Results

Page 3: "Our screens for putative sex-linked markers using the three strategies yielded signatures that matched our expectations for either an XY or a ZW system in all datasets (Tables S1 and S2), indicating the prevalence of false positives." I do not understand what the authors mean by that. Which "3 strategies"? Those mentioned in figure S1 and S2? I see they are mentioned in the Material and Methods section. This should be better presented here.

Page 3: "All male and female individuals showed heterozygosity and homozygosity at their polymorphic loci". "Respectively" should be added.

Page 4: "By indirect mapping, we compared the sex-linked markers of all nine Amolops species to the P. adspersus genome using L. catesbeianus". This should be developed. Why do the authors use this method and what does mean "using L. castebeianus"? used for what. Even if the authors describe nicely the reasoning in the Methods section, that would be appropriate to have a bit of an explanation here.

Page 4: "The reasons for the unsuccessful alignments of sex-linked mark-ers were the high divergence time between the Amolops species and the third reference frame". I do not understand this statement. How do the authors come to this conclusion? And if that is the case, why did they use these third party reference genomes in the first place?

Page 4: "This indicates that these two genes are located on different chromosomes rather than on identical ones." In which animals?

Page 5: "there is no shared XY site (i.e., sex-linked locus) except one between alternate species pairs (the site is shown in Fig. 1)". While in the legend of figure 1 it is written: "there is no shared XY site shown". I do not understand. Also, some dots are connected to Cyp19d1 and Ar genes. Why? I also have problems to understand how the dots are distributed along the dotted lines in the figure. Is there a logic behind it? This should be clarified.

Page 5: "Using this site to construct a multispecies gene tree, the results are shown in Fig. 2, which are consistent with the prediction in Fig. 3a (See section 4.9 of Ma-terials and Methods)." The authors should better describe their results in the text. If I understand well, Figure 3 is sort of a model (two different scenarios). This should be presented first and explained as it is part of the reasoning itself leading to a conclusion.

Page 6: "As shown in Fig. 4, the results were in line with the predictions shown in Fig. 3b". Same as for my previous comment.

Page 5-8: I find the logical flow of the section 2.6 very hard to grasp.

Discussion

Page 8: "the trigger gene probably changed between the two pairs of sex chromosomes (Fig. 1)". I do not understand this.

Page 8: "Across closed species of a lineage restricted in a region". Geographical region I presume?

Page 9: "Using similar approaches, the GBS markers in this study might have also failed to explore the small ancestral SDR among the eight torrent frogs." I am not sure to understand the meaning of this, especially the use of "explore" in this context.

Page 9: "indicating that Xs and Ys should have con-tinued to recombine". What are Xs and Ys?

Conclusion

What do the authors conclude from their work? This is, somehow, missing.

Reviewer 2 Report

The ms reports a very interesting study on sex-linked markers of Amolops species. I point out below some specific comments to be considered by the authors.

1. Why do you use RAD-seq markers for 2 of the analyzed species? How do you compare RAD-seq and GBS with respect to their performance in the prospection of sex-linked markers?

2. Abstract – “… sex chromosomes have been described as homomorphic.” – In addition to homomorphic sex chromosomes, some heteromorphic sex chromosomes have also been described. Please, rephrase this sentence to avoid any misinterpretation.

3. Abstract – “Here, we examined the patterns of sex chromosome evolution” – It sounds a little overrated. Some aspects of the sex chromosome of Amolops spp. were revealed, but this is not sufficient to assess the processes involved in sex chromosome evolution.

4. Abstract –“ We found that sex-chromosome homomorphy results…” – I don’t think your results prove that the analyzed species have homomorphic sex chromosomes (although they do not suggest the contrary). How do you exclude the possibility that the prospected sex-linked markers are in a pseudo-autosomal region? Are you sure all the sex-linked loci were found?

Usually, the classification of sex chromosomes as heteromorphic or homomorphic is based on cytogenetic-level analysis. Is there any cytogenetic information about the karyotypes of interest? Unless a chromosome-level genome assembly or a classical cytogenetic analysis is available, I would avoid classifying the sex chromosomes in focus as homomorphic.   

5. Abstract – “A lower turnover rate of non-homologous sex chromosomes exists in these torrent frogs.” – Replace “lower” with “low”.

6. Appendix 3 – Please, inform the meaning of M and n.

7. Page 3 – “All male and female individuals showed heterozygosity and homozygosity at their polymorphic loci (Appendix 4).” – I suggest a revision of this sentence. Please, consider something like this: “All male individuals were homozygous, while all females were heterozygous for…”

8. Page 3 – “Thus, the XY sex determination system was further verified by PCR amplification in all nine Amolops species (Table S3).” – Verified or confirmed?

9. Figure 1 - What do the colored bars in the chromosome ideograms indicate? Do they indicate the relative position of the sex-SNPs of different species? I don’t think so, but the representation allows such interpretation.

10. Figure S2. As this figure is related to Figure 1, it would be easier for the reader if the SNPs/loci of a species were shown in the same color in figures 1 and S2. For example, as the sex-SNPs of A. montzorum are shown in green in Figure 1, the red arrows in Figure S2 would be colored in green (instead of red).

11. Figure 1/legend: “Colored dots show sex-linked SNPs of nine Amolops species, and there is no shared XY site shown.” – The highlighted text should be in another sentence and should be better explained.

12. Page 5: “The sex chromosome is mapped with the sex-linked markers in each of the nine Amolops species, and there is no shared XY site (i.e., sex-linked locus) except one between alternate species pairs (the site is shown in Fig. 1).” – This sentence is unclear. Which sex-linked locus is shared? Which are the “alternate species pairs”?

13. Figure 2 – legend: “Sequences from males show in blue, and those from females in red.” – Replace with “Sequences from males are shown in blue, and those from females in red. 

14. Figure 2 – legend: “Bootstrap values are shown for the main branches, when above 50%.” I couldn’t see any bootstrap value in this figure.

15. Page 5 – “All male genotypes for this sex-linked locus were T/A heterozygous, whereas all female genotypes were T/T homozygous, which confirmed that the marker was gender-diagnostic in both species (Table S5). Using this site to construct a multispecies gene tree, the results are shown in Fig. 2, which are consistent with the prediction in Fig. 3a (See section 4.9 of Materials and Methods).” – I apologize to the authors, but I did not understand how many markers were used in the RAxML analysis that resulted in this tree. If only one SNP was considered (as suggested by “Using this site to construct a multispecies tree”), why did you run a RAxML analysis?

16. Page 6 – “As shown in Fig. 4, the results were in line with the predictions shown in Fig. 3b.” – Which results? Please, open this paragraph with a brief comment about the data you will report.

17. Page 6 – “All five sex-linked loci of A. chunganensis constitute a species genealogy, with the X gamete and Y gamete forming clades that are sister to one another, whereas the cross-species homologous sequences of the remaining species constitute a species genealogy.” – I found this sentence too weird. Did you mean “All the X-specific sequences of A. chunganensis were clustered together and apart from the Y-specific sequences of this species. In contrast, the X- and Y-specific sequences of each remaining species were not clustered in distinct groups.”?

18. Page 7 – “This indicated that the remaining seven species were no longer sex-linked” – Sex-linked species? Please, revise this sentence.

Round 2

Reviewer 1 Report

The authors addressed all my previous comments carefully and I find the manuscript much clearer in the present version. The conclusion could have highlighted a bit better the contribution of the findings to the field, but I guess it is fine enough as it is. I just have minor comments that the authors should take into consideration before their final submission.

Page 2: " The early cytogenetic studies show that all investigated Amolops species are homomorphic sex chromosome, except one species (A. mantzorum) have morphologically distinct sex chromosomes." Something wrong in the sentence. I guess it is "one species that has morpholocally…"

Page 3, line 100: " appears to have the lower turnover rate". Should be either "a lower" or "the lowest".

Page 4, line 197-198: " This high sequence divergence could be caused by that these sex-linked markers is non-coding genes". "is" should be "are".
